

# The burgeoning spatial multi-omics in human gastrointestinal cancers

Weizheng Liang[1,*], Zhenpeng Zhu[2,3,*], Dandan Xu[1], Peng Wang[2,3], Fei Guo[2], Haoshan Xiao[2,3], Chenyang Hou[2,3], Jun Xue[2], Xuejun Zhi[4] and Rensen Ran[1,5]

[1] Central Laboratory, The First Affiliated Hospital of Hebei North University, Zhangjiakou, Hebei province, China

[2] Department of Surgery, The First Affiliated Hospital of Hebei North University, Zhangjiakou, Hebei Province, China

[3] Hebei North University, Zhangjiakou, Hebei Province, China

[4] Department of Respiratory and Critical Care Medicine, The First Affiliated Hospital of Hebei North University, Zhangjiakou, Hebei province, China

[5] Department of Chemical Biology, School of Life Sciences, Southern University of Science and Technology, Shenzhen, China

* These authors contributed equally to this work.

Corresponding authors
Xuejun Zhi, zgzhixj@sohu.com
Rensen Ran, sankeshumy@163.com

## ABSTRACT

The development and progression of diseases in multicellular organisms unfold within the intricate three-dimensional body environment. Thus, to comprehensively understand the molecular mechanisms governing individual development and disease progression, precise acquisition of biological data, including genome, transcriptome, proteome, metabolome, and epigenome, with single-cell resolution and spatial information within the body's three-dimensional context, is essential. This foundational information serves as the basis for deciphering cellular and molecular mechanisms. Although single-cell multi-omics technology can provide biological information such as genome, transcriptome, proteome, metabolome, and epigenome with single-cell resolution, the sample preparation process leads to the loss of spatial information. Spatial multi-omics technology, however, facilitates the characterization of biological data, such as genome, transcriptome, proteome, metabolome, and epigenome in tissue samples, while retaining their spatial context. Consequently, these techniques significantly enhance our understanding of individual development and disease pathology. Currently, spatial multi-omics technology has played a vital role in elucidating various processes in tumor biology, including tumor occurrence, development, and metastasis, particularly in the realms of tumor immunity and the heterogeneity of the tumor microenvironment. Therefore, this article provides a comprehensive overview of spatial transcriptomics, spatial proteomics, and spatial metabolomics-related technologies and their application in research concerning esophageal cancer, gastric cancer, and colorectal cancer. The objective is to foster the research and implementation of spatial multi-omics technology in digestive tumor diseases. This review will provide new technical insights for molecular biology researchers.

# INTRODUCTION

In multicellular organisms, the processes of development and disease progression transpire within intricate three-dimensional *in vivo* environments. Each cell within these organisms inhabits a microenvironment that is intricately linked to its cellular fate, and these microenvironments exhibit pronounced heterogeneity. Thus, to comprehensively fathom the molecular underpinnings of individual development and the onset and progression of diseases lies in acquiring precise biological data encompassing the spatial coordinates of the cell's genome, transcriptome, proteome, metabolome, and epigenome within the *in vivo* three-dimensional milieu. This method facilitates the exploration of both microenvironment-cell interactions and the intricate regulatory dynamics of cellular functions, thereby affording insight into the pertinent molecular mechanisms. Notably, while conventional single-cell multi-omics techniques result in the loss of cell spatial coordinate data during sample preparation, recent advances in spatial multi-omics methodologies have enabled the retention of this information (*Badia-i-Mompel et al., 2023*; *Velten & Stegle, 2023*; *Bressan, Battistoni & Hannon, 2023*). Consequently, the synergistic amalgamation of single-cell multi-omics and spatial multi-omics techniques holds promise for the comprehensive analysis of biological information, encompassing the genome, transcriptome, proteome, metabolome, and epigenome, while retaining essential spatial information.

Digestive tract tumors, a prevalent form of cancer worldwide, contribute significantly to annual mortality rates (*Hirata et al., 2023*; *Thrift, Wenker & El-Serag, 2023*). While single-cell multi-omics techniques have been extensively employed in recent years to elucidate the intricate molecular mechanisms and heterogeneity associated with digestive tract tumor development, a comprehensive grasp of biological information, which includes spatial coordinate data for cell genomes, transcriptomes, proteomes, metabolomes, and epigenomes, necessitates the application of research methodologies such as spatial multi-omics techniques (*Li et al., 2023a*; *Frank et al., 2021*). The integration of single-cell multi-omics and spatial multi-omics approaches enables the acquisition of valuable insights into the molecular underpinnings of tumor cell development and heterogeneity within the three-dimensional microenvironment (*Badia-i-Mompel et al., 2023*; *Velten & Stegle, 2023*; *Bressan, Battistoni & Hannon, 2023*; *Baysoy et al., 2023*). Consequently, this article offers a comprehensive review of spatial multi-omics techniques and their relevance in the context of digestive tract tumor diseases, with the objective of promoting their utilization in digestive tract tumor research (Fig. 1). Furthermore, it is hoped that forthcoming research endeavors will harness spatial multi-omics techniques to achieve a more comprehensive and precise understanding of the molecular mechanisms and heterogeneity that underlie the development of digestive tract tumor diseases. Here, this article can provide insights into cutting-edge technology for researchers engaged in molecular research, and provide some new ideas for researchers engaged in biomedical and tumor research.

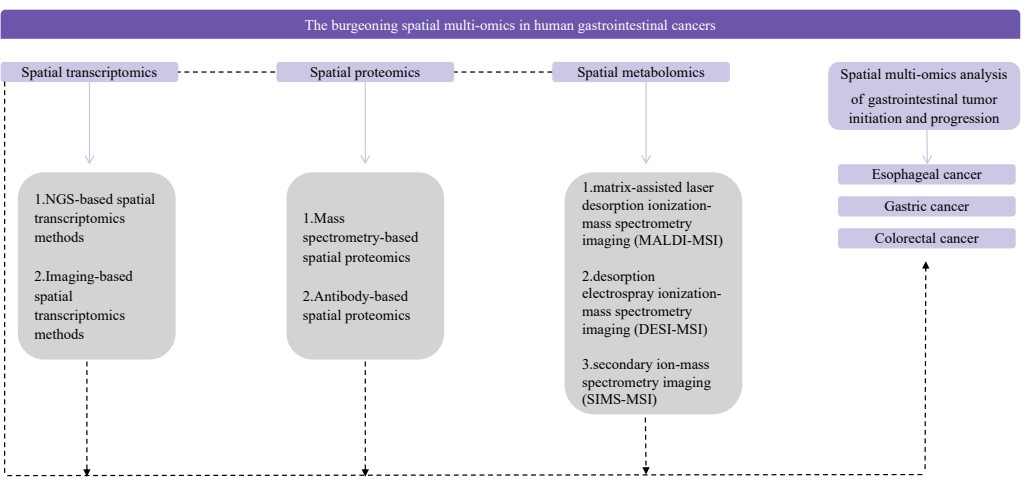

**Figure 1** Flow chart of the review.

# SURVEY METHODOLOGY

We used specialized databases including PubMed and Web of Science for the literature review. This review explores the current status of multi-omics technologies including spatial transcriptomics, spatial proteomics, and spatial metabolomics, with a focus on the application of multi-omics technologies in digestive tract tumors. This review will provide the latest technical insights and study design ideas for our molecular researchers.

## Necessity and advantages of spatial multi-omics development

Single-cell sequencing technology is also a new generation of high-throughput sequencing technology, including single-cell genome sequencing, transcriptome sequencing, epigenome sequencing, proteome sequencing, and multi-omics sequencing. Compared with traditional omics, the combination of single-cell and multi-omics technologies has the advantage that the heterogeneity of cells can be observed at the single cell level, and the target molecules can be lowered from multicellular populations to single cell molecules. Biological processes occur in a spatial context, and the three-dimensional (3D) arrangement of cells in tissues has a profound impact on their functions, such as limiting cell-to-cell interactions by modulating contact or short-range paracrine signals. Although the technical development of single-cell multi-omics makes today's tumor research has reached the subcellular dimension, compared with the booming development of spatial multi-omics, single-cell multi-omics seems to discard the spatial dimension information between cells, and the development of spatial multi-omics just makes up for these shortcomings (Fig. 2) (*Vandereyken et al., 2023*; *Bressan, Battistoni & Hannon, 2023*). Single-cell multi-omics technology mainly emphasizes cell grouping, which often loses the phenomenon of cell location information. The missed spatial location information is also the loss of spatial heterogeneity, and can not further correlate the molecular mechanism with the tissue *in situ*. This misses strong spatial evidence for the study of cell-to-cell interactions, primitive gene expression, microenvironment between tissues,

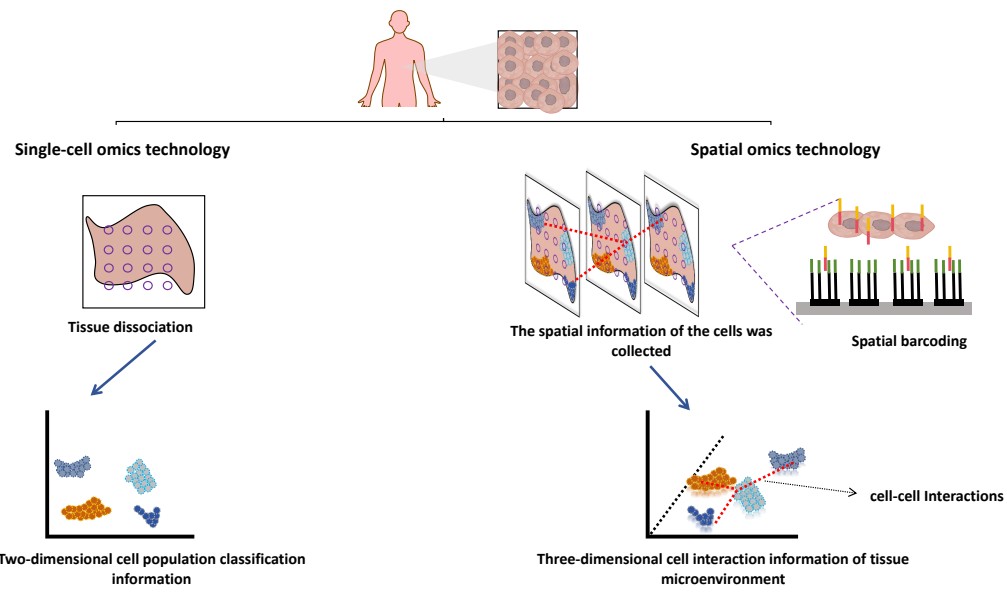

**Figure 2** **Spatial multi-omics techniques.** Created with Figdraw, https://www.figdraw.com.

and tissue growth (*Hsieh et al., 2022*). The body is a complex and three-dimensional, and the functional state between cells is closely related to the physical and endocrine regulation between adjacent cells, which is extra important for the study of the location and spatial information of single cells (*Li et al., 2022*). However, structure determines function, and the complex structure of cells contributes to their different functions. Different cells are different in time and space. The interaction between cells and their spatial location and structural information form a unique cellular microenvironment to maintain the stability of the complex functions of the human body (*Wang & Fan, 2021*). Visualization of cellular spatial heterogeneity and spatial structure of tumor microenvironment, which cannot be addressed by single-cell multi-omics technology (*Walsh & Quail, 2023*).

## Spatial transcriptomics

Spatial transcriptomics is a technology designed to retain the spatial context of tissues while concurrently elucidating the transcriptomic profiles of tissue sections. This approach allows for the precise mapping of gene expression within tissue sections, unveiling the spatial distribution of distinct cell types, their intercellular interactions, and gene expression profiles across diverse tissue regions (*Moor & Itzkovitz, 2017*). Such insights are invaluable for comprehending the molecular underpinnings of developmental processes and disease etiology and progression. Spatial transcriptomics can be categorized into two major types: those rooted in next-generation sequencing (NGS) methods and those relying on imaging techniques.

## NGS-based spatial transcriptomics methods

Spatial transcriptomics technology, initially introduced in 2016, hinges on barcoded glass slides to furnish spatial localization information for RNA transcripts within tissue

sections (*Stahl et al., 2016*). This information is subsequently employed to reconstruct the transcriptomic landscape within the tissue's three-dimensional architecture. Essentially, spatial transcriptomics technology employs specialized glass slides with an array of discreet microwells, each covering one or more oligonucleotide chains. These oligonucleotide chains incorporate a spatial barcode specific to each microwell, along with a poly T tail that complements the poly A tail of mRNA, enabling efficient capture. Following enzymatic permeabilization, mRNAs from tissue sections are liberated from cells and entrapped by the poly T tails within the microwells. Subsequently, cDNA synthesis takes place on the glass slide, followed by sequencing using NGS (Fig. 3A). This procedure facilitates the retrieval of the original spatial location information for mRNA, based on spatial barcodes.

In late 2018, 10x Genomics advanced the field of spatial transcriptomics with the introduction of their Visium spatial transcriptomics technology, which enhanced both resolution and operational efficiency (*Marx, 2021*). Subsequently, in 2019, *Vickovic et al. (2019)* presented the High-Definition Spatial Transcriptomics (HDST) technology. HDST outperformed traditional spatial transcriptomics methods by employing an organized magnetic bead array and a split-pool approach to generate high-resolution (2 μm) and high-density (several million) bead arrays. Simultaneously, the Slide-seq technology emerged, capturing mRNAs in frozen tissue sections at a spatial resolution of 10 μm by placing random barcode-coated magnetic beads on glass slides, achieving single-cell-sized resolution but requiring supplemental single-cell RNA sequencing (scRNA-seq) data for heightened sensitivity (*Rodriques et al., 2019*). Subsequent to this, a Slide-seqV2 technology was developed, nearing single-cell resolution, with enhancements in bead synthesis, library generation, and sequencing processes (*Stickels et al., 2021*). This improved version of Slide-seq enhanced mRNA capture efficiency approximately tenfold compared to its predecessor, albeit requiring further development for commercialization. Also, in 2019, NanoString introduced the GeoMx Digital Spatial Profiler (DSP) technology, tailored for spatial multi-omics analysis in the context of tumor immunity and the tumor microenvironment (*Hernandez et al., 2022*). This technology quantifies the number and spatial distribution of various immune cell-related protein markers in the tumor microenvironment. It enables region-specific selection and precision laser cleavage of DNA oligos linking antibodies or RNA probes, thus releasing DNA oligos for subsequent quantification. Notably, its advantage lies in the precision of laser activation, allowing for resolutions down to the single-cell level. However, it should be noted that its drawback is the requirement for specially designed antibodies and probes.

In 2020, the Rong Fan team developed a method known as Deterministic Barcoding in Tissue for spatial omics sequencing (DBiT-seq) for tissue section analysis (*Liu et al., 2020*). DBiT-seq employs microfluidic channels to transport barcode probes, achieving a 10 μm resolution and enabling concurrent capture of transcripts and chosen protein targets. Unlike complex imaging methods, it utilizes high-throughput NGS and DNA barcoding to concurrently acquire RNA transcriptome and proteomics data, enhancing sample throughput and cost-effectiveness. DBiT-seq represents a novel spatial omics technology that is facilitated by simple equipment and user-friendly for researchers. In 2021, Jun Hee Lee's research team at the University of Michigan Medical School utilized

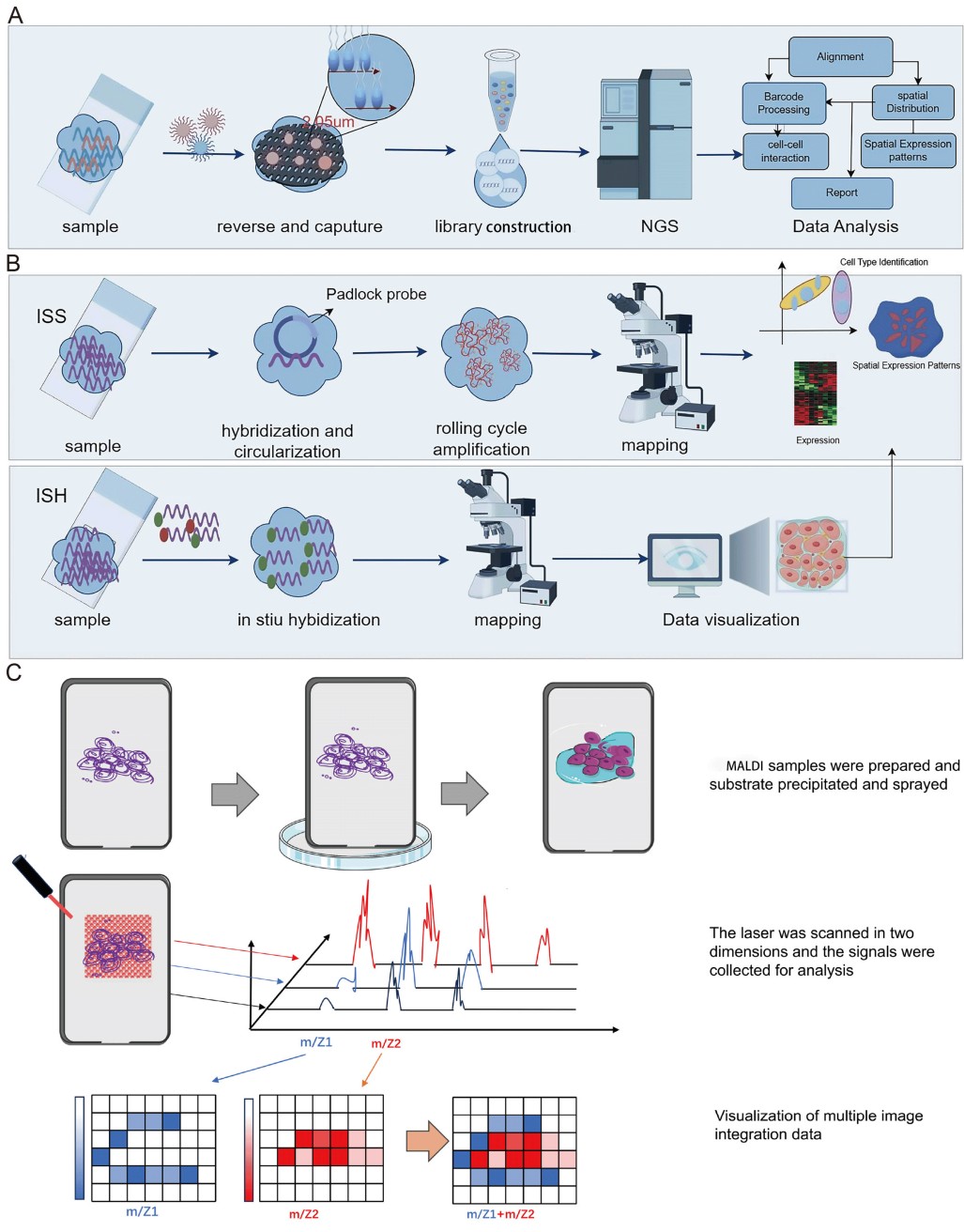

**Figure 3** **Schematic diagrams of spatial transcriptomic and matrix-assisted laser desorption/ion-ization (MALDI).** (A) Procedures of NGS-based method of spatial transcriptomic. (B) Procedures of imaging-based method of spatial transcriptomic. (C) Schematic diagram of MALDI. The process begins with the preparation of the MALDI sample, followed by the application of MALDI for the matrix deposition. Subsequently, laser scanning technology is employed for the collection, processing, and analysis of signals. Finally, the signals and multi-image data are integrated for visualization. Created with Figdraw, https://www.figdraw.com.

spatial barcode technology and Illumina sequencing to achieve sub-micron resolution in spatial transcriptomics, denoting the method as Seq-Scope (*Cho et al., 2021*). It effectively visualized transcriptome variations at cellular and subcellular levels across various tissues. In 2022, BGI Group, collaborating with multiple institutions, developed Stereo-seq, a high-field-of-view nanoscale resolution spatial omics technology (*Chen et al., 2022*). This innovation permits high-throughput transcriptome analysis of tissue sections at subcellular resolution, with scalability up to centimeter-level areas. Stereo-seq is distinguished by its heightened sensitivity and uniform capture rates. It has the potential to complement medical imaging and histopathological data, potentially becoming a specialized diagnostic tool in routine clinical practice.

In essence, contemporary NGS-based spatial transcriptomics methods predominantly entail the retrieval of spatially barcoded RNA for subsequent sequencing. These techniques excel in their applicability to large-scale tissue section sequencing. Nonetheless, they fall short in achieving precise single-cell resolution, and RNA detection efficiency remains comparatively limited.

## Imaging-based spatial transcriptomics methods

Spatial transcriptomics technology, grounded in imaging methodologies, primarily encompasses two modalities: *In Situ* Sequencing (ISS) and *In Situ* Hybridization (ISH) (Fig. 3B). The ISS method relies exclusively on ligase enzymes for the concatenation of two DNA segments, specifically a primer with a predefined sequence and a probe congruent with the template (*Ke et al., 2013*). In 2013, *Ke et al. (2013)* developed the first ISS method, which used padlock probes to target known genes. Essentially, the ISS method involves reverse transcribing mRNA into cDNA within intact tissue slices, and then binding padlock probes to the cDNA. This allows for high throughput when subjected to bulk sequencing but sacrifices spatial resolution. The advantages of the ISS method mainly include single-cell resolution and subcellular transcript localization, while the drawback is primarily lower detection efficiency (*Moses & Pachter, 2022*). In 2015, *Lee et al. (2015)* developed Fluorescence *In situ* Sequencing (FISSEQ), a non-targeted method capturing all RNA types. FISSEQ substantially augmented the detection throughput of *in situ* sequencing, enabling the acquisition of gene expression maps across the entire genome, including gene expression, RNA splicing, and post-transcriptional modifications while retaining spatial positional information. However, FISSEQ's limitation is evident in its significantly lower gene yield compared to RNA-seq, providing only about 200 mRNA fragments per cell, whereas single-cell RNA-seq yields approximately 40,000 mRNA fragments (*Lee et al., 2015*). Consequently, the FISSEQ method has lower sequencing depth, potentially missing low-abundance RNA transcripts, thereby failing to deliver comprehensive information about intracellular RNA. In 2018, BaristaSeq technology was reported, which increased amplification efficiency by fivefold and achieved a sequencing accuracy of at least 97% (*Chen et al., 2018*). Subsequently, STARmap technology was developed, eliminating the need for cDNA conversion and reducing noise by introducing a second hybridization step (*Wang et al., 2018*). Moreover, it obviates the requirement for tissue clearing, leading to heightened

sensitivity. As such, this technology exhibits promise in enabling gene expression detection at single-cell resolution within three-dimensional space.

Spatial transcriptomics technology, rooted in ISH methods, employs labeled nucleic acid probes to precisely ascertain the spatial distribution and concentration of DNA and RNA within biological tissues and cells, such as single-molecule fluorescence *in situ* hybridization (smFISH) technology (*Femino et al., 1998*). In smFISH, fluorescently tagged probes are employed to selectively hybridize with target RNA molecules, generating distinct fluorescent signals. Subsequently, microscopy facilitates the visualization of RNA's spatial positioning and quantification. Nevertheless, it is imperative to note that smFISH is constrained in its capacity to simultaneously target only a limited number of genes. In 2014, the Sequential Fluorescence *In Situ* Hybridization (SeqFISH) technology was developed (*Coskun & Cai, 2016*). SeqFISH is a multi-round smFISH method, which has the drawbacks of being high in cost and time-consuming. Subsequently, in 2015, Professor Xiaowei Zhuang of Harvard University introduced the MERFISH (Multiplexed Error-Robust Fluorescence *In situ* Hybridization) technology (*Moffitt & Zhuang, 2016*). This technique enables the concurrent assessment of expression levels and spatial distribution of thousands of RNA species within a single cell. In simple terms, MERFISH technology leverages a combination of labeling and continuous imaging techniques to enhance detection throughput. It utilizes binary barcoding to rectify errors in single-molecule labeling and detection. By conducting multiple rounds of imaging, MERFISH simultaneously deciphers numerous distinct barcodes, enabling the comprehensive assessment of the expression levels and spatial distribution of thousands of RNA species. In 2019, Professor Long Cai developed the seqFISH+ technology, which shares core principles with MERFISH (*Eng et al., 2019*). SeqFISH+ employs a four-round imaging encoding approach and expands the fluorescence imaging channels from a single laser to three (640 nm, 561 nm, 488 nm), substantially increasing the coding capacity.

In conclusion, as deduced from the above, it is evident that the merits of ISS technology encompass single-cell resolution, subcellular transcript localization, and applicability to larger tissue regions. However, it is associated with the limitation of relatively lower detection efficiency. On the other hand, ISH-based methods substantially expand the detectable area, but they also encounter certain challenges. For instance, techniques like smFISH struggle to isolate individual cells from complex backgrounds involving issues such as signal interference and transcript accumulation.

## Spatial proteomics

Proteomics is the systematic investigation of all protein structures and functions expressed within cellular or tissue genomes, particularly under specific environmental or temporal conditions. It employs high-resolution mass spectrometry and advanced bioinformatics to unravel the mechanistic intricacies of physiological or pathological changes. Spatial proteomics technology combines highly sensitive mass spectrometry and ultra-high-resolution microscopy, complemented by cellular phenotyping, enabling precise protein localization and functional analysis within cells and tissues (*Vandereyken et al., 2023*; *Mund, Brunner & Mann, 2022*; *Bahrami et al., 2023*; *Eisenstein, 2022*). Regarding research

![PeerJ]

methodologies, spatial proteomics technology falls into two primary categories: mass spectrometry-based and antibody-based spatial proteomics.

## Mass spectrometry-based spatial proteomics

Mass spectrometry imaging (MSI) is an exceptionally sensitive molecular imaging method that originates from mass spectrometry. It facilitates the direct identification and spatial localization of proteins within tissue sections, single cells, or various material surfaces, thus furnishing proteomic insights with spatial resolution (*Chaurand, Stoeckli & Caprioli, 1999*; *Chaurand, Schwartz & Caprioli, 2004*). Prominent attributes of MSI research encompass its remarkable sensitivity, label-free peptide and protein imaging capacity, and spatial resolution spanning from the individual to cellular scales. Additionally, MSI permits the concurrent imaging of numerous distinct molecules within a single experiment. The most common MSI technique is matrix assisted laser desorption ionization-mass spectrometry imaging (MALDI-MSI), characterized by a mechanism where a matrix absorbs laser energy, facilitating ionization of sample molecules (*Ryan, Spraggins & Caprioli, 2019*; *Stoeckli et al., 2001*). The ionized target analytes are subsequently introduced into a mass spectrometer for identification. Simultaneously, raster scanning enables the generation of tissue images. MALDI-MSI mass spectrometry tissue *in situ* imaging can analyze numerous protein or peptide fragments at a resolution in the order of tens of micrometers (Fig. 3C). However, this approach commonly relies on hematoxylin and eosin staining to identify regions of interest (ROIs) and is constrained by limited resolution (Fig. 3C), posing challenges for precise analysis of tumor microenvironment substructures like tumor-infiltrating lymphocytes, tumor-associated macrophages, and tertiary lymphoid structures. The Multiplexed Ion Beam Imaging (MIBI) technology, as reported by Garry Nolan and his team in 2014, can be distilled as an antibody-based immune reaction utilizing distinct isotopic labels for individual antibodies (*Angelo et al., 2014*; *Rost et al., 2017*; *Liu et al., 2022b*; *Liu et al., 2022a*). This allows for targeted capture and analysis, thus providing spatial information on multiple proteins (over 40 commercially available targets) at the single-cell level (Fig. 4A). For instance, MIBI technology allows the simultaneous detection of 36 representative proteins in triple-negative breast cancer tumor and immune cells, providing a comprehensive depiction of their spatial distribution within breast cancer tissue (*Keren et al., 2018*). However, the extensive adoption of MIBI technology is impeded by its relatively high hardware costs, the expense associated with isotopically labeled antibodies, and the need for specific tissue fixation matrices and customizable isotopic selections. The subsequently developed Multiplexed Ion Beam Imaging-Time of Flight (MIBI-TOF) technology offers an exceptional resolution of 260 nm and near-single-molecule sensitivity (*Keren et al., 2019*; *Baharlou et al., 2019*; *Keren et al., 2018*) (Fig. 4A). This technology enables routine and robust imaging of formalin-fixed paraffin-embedded (FFPE) tissue samples but is reliant on isotopically labeled antibodies (Fig. 4A). Various other mass spectrometry-based spatial proteomics techniques have been developed, each with distinct attributes and limitations (*Lundberg & Borner, 2019*). In summary, mass spectrometry-based spatial proteomics technology is rapidly advancing, with increased sensitivity, the ability to detect more targets, and a higher degree of precision in single-cell

resolution. It is highly likely that future developments will yield mass spectrometry-based spatial proteomics technologies meeting these criteria.

## Antibody-based spatial proteomics

The Co-Detection by Indexing (CODEX) technology is based on antibody-based spatial proteomics technology. Simply put, it involves the individual labeling of antibodies with distinct oligonucleotide barcode tags, followed by the selective detection and binding of the fluorescent dye of the "secondary antibody" to the complementary oligonucleotide sequence (Fig. 4B). This technological advancement overcomes the constraint of the count of visible light spectral fluorescence imaging channels, effortlessly achieving the simultaneous detection and analysis of 50 or more proteins (*Goltsev et al., 2018*; *Black et al., 2021*; *Kuswanto, Nolan & Lu, 2023*; *Hickey et al., 2021*). In simple terms, the operational process involves initially mixing several dozen antibodies, each with distinct barcode labels, and incubating them collectively with the tissue section following a standard primary antibody mixing procedure. Upon binding each specific antigen on the tissue section, a fluorescent dye distinguishes the antibodies marked with different barcodes. This approach effectively assigns distinct oligonucleotide barcode labels to individual antigen targets *via* antibody recognition. Subsequently, the fluorescence imaging is carried out in stages, with each round detecting three targets, incorporating three-color fluorescent label reporters that recognize and bind to their corresponding barcodes. This facilitates the attachment of fluorescent dyes to specific antibodies and antigens, subsequently recorded *via* imaging. The detachable fluorescently labeled reporter is then gently washed, and a three-color fluorescent label reporter recognizing other barcodes is introduced for coloration imaging (Fig. 4B). This process iterates until all targets are identified. Ultimately, the superimposed images showcase the spatial distribution and interrelationships of up to 50 distinct targets within the same sample slice (*Black et al., 2021*; *Wang et al., 2021a*). Overall, the CODEX technology not only comprehensively examines intricate spatial data within tissue samples but also identifies fresh frozen samples, paraffin-embedded tissue samples and other cellular samples, preserving sample integrity.

DSP is also a technology that can be used for spatial transcriptomics or proteomics research. Simply put, it operates through probe *in situ* hybridization and antigen-antibody interaction, enabling the identification of gene and protein expression within tissue sections (Fig. 4C). Photocleavable linkers are utilized to affix oligonucleotide tag sequences (DSP barcodes) to the probes and antibodies, facilitating high-throughput detection and quantification of genes and proteins (Fig. 4C). The most prominent feature of DSP technology lies in its "targeting" capability. Specifically, DSP can target specific gene/protein expression characteristics in the target areas of tissue sections, thereby narrowing the scope of research and significantly enhancing research efficiency. This targeting ability is fundamentally underpinned by the amalgamation of DSP with multiple immunofluorescence techniques. By implementing multiple immunofluorescent staining, the structural attributes of tissue slices can be predetermined, enabling researchers to methodically sequence and scrutinize target regions in alignment with their investigative objectives (Fig. 4C). This deliberate identification of targeted information effectively

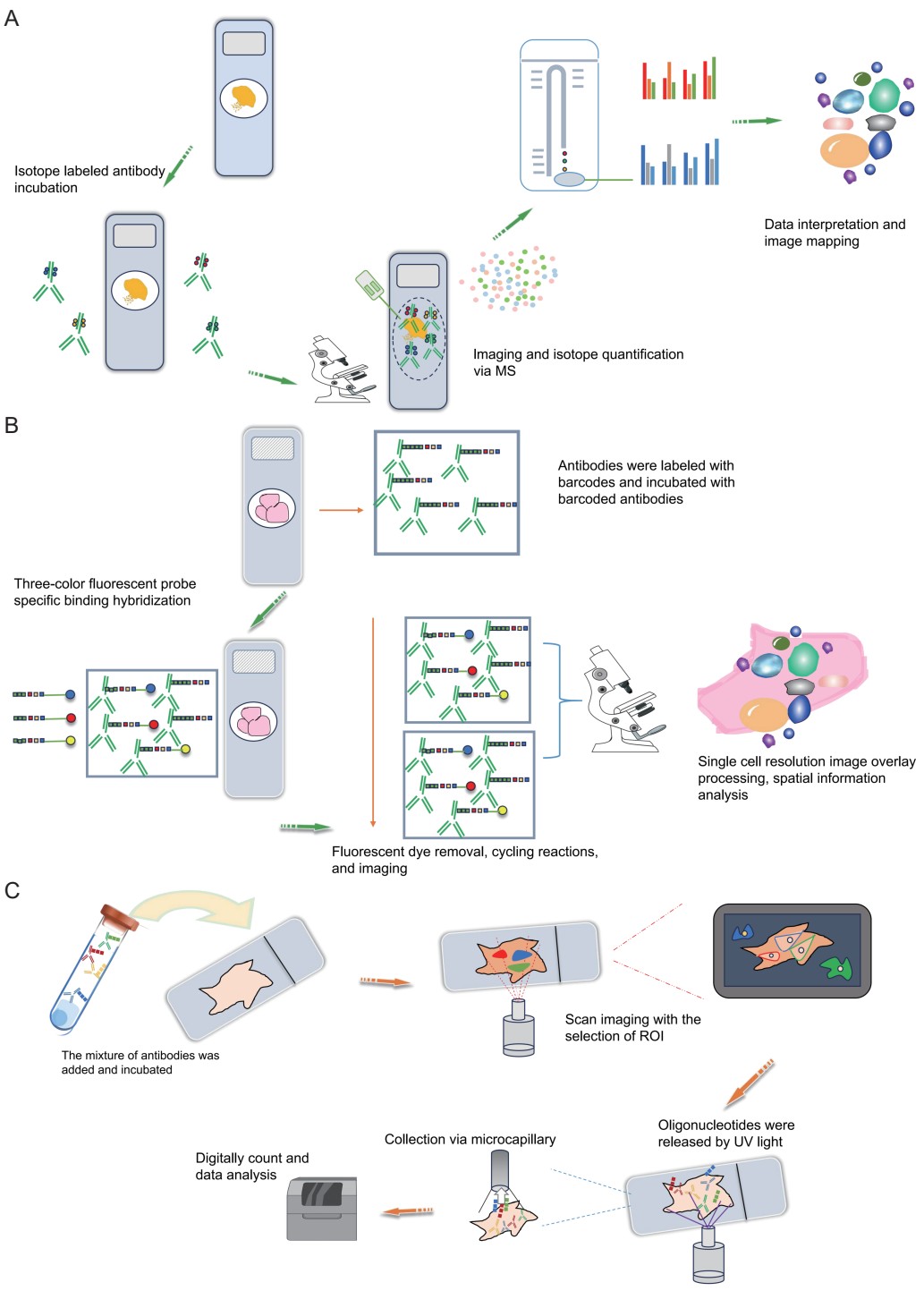

**Figure 4** **Schematic diagrams of MIBI, MIBI and DSP.** (A) Schematic diagram of MIBI. Commence by incubating the sample with isotope labeling and antibodies. Subsequently, conduct imaging and isotope quantification using mass spectrometry (MS). Finally, integrate and analyze the data to elucidate spatial organization at the single-cell resolution. (continued on next page...)

**Figure 4 (…continued)**
(B) Schematic diagram of CODEX. To begin, conjugate antibodies with barcode tags and incubate them with the sample along with fluorescent tricolor probes for specific hybridization. Next, remove excess fluorescent dyes, and proceed to the cycling reaction and imaging steps. Finally, overlay the images at single-cell resolution and conduct spatial information analysis. (C) Schematic diagram of DSP. Initially, incubate the sample with a mixed antibody population. Subsequently, designate specific regions of interest (ROIs) for scanning imaging, succeeded by UV light exposure to liberate oligonucleotides. Finally, utilize microcapillaries for collection and integrate the data for comprehensive analysis. Created with Figdraw, https://www.figdraw.com.

circumvents the challenges associated with unselective data mining across the entirety of the section. This aspect constitutes the key distinction between DSP technology and the 10X Visium spatial transcriptomics technology. At present, DSP technology facilitates spatial proteins detection across key translational medical domains, encompassing immunology, tumor immunology, and neuroscience, in addition to targeting common drug targets and signaling pathways, with the potential to detect up to 100 specific protein targets (*Hernandez et al., 2022*; *Merritt et al., 2020*). DSP can be synergistically integrated with scRNA-seq to investigate the cellular distribution patterns within the tumor immune microenvironment, enabling an intricate analysis of both the gene expression profiles of tumor cells and the surrounding microenvironment. This integrated approach aids in identifying prospective therapeutic targets for tumors, thereby holding substantial implications for the advancement of precise tumor therapy (*Jerby-Arnon et al., 2021*). The landscape of various antibody-based spatial proteomics technologies is rapidly progressing, evident in the growing use of diverse antibodies for detection and their integration with other multi-omics methodologies, representing an anticipated trajectory for future development.

## Spatial metabolomics

Metabolomics is a comprehensive and quantitative analytical technology that examines the dynamic variations in the metabolic profiles of organisms in response to specific stimuli (*Fiehn, 2002*). It primarily focuses on small-molecule metabolites weighing less than 1,000 Daltons (Da), including sugars, organic acids, lipids, amino acids, and aromatic hydrocarbons. Consequently, metabolomics represents an emerging field of microscopic inquiry, succeeding the trajectories of proteomics and genomics. Mass spectrometry is a reliable method for elucidating metabolic products, offering a comprehensive portrayal of the metabolic landscape during the analysis of cancer cell metabolites and cancer metabolic reprogramming pathways (*Kowalczyk et al., 2020*; *Ciocan-Cartita et al., 2019*). Nevertheless, conventional mass spectrometry approaches can compromise the spatial information of molecules due to sample preparation steps. Consequently, spatial metabolomics technology has emerged to address this limitation (*Taylor, Lukowski & Anderton, 2021*; *Alexandrov, 2020*; *Alexandrov, 2023*; *Saunders et al., 2023*). Spatial metabolomics technology is an emerging research methodology that integrates diverse technical platforms and data sources for high-throughput and precise quantitative analysis of intracellular metabolites (*Planque et al., 2023*; *Saunders et al., 2023*). It also examines the spatial distribution of these metabolites to investigate their interactions

and reciprocal regulatory mechanisms. Simply put, spatial metabolomics represents an innovative molecular imaging method founded on mass spectrometry imaging (MSI) and high-throughput sequencing. This method involves directly applying samples onto glass slides or using nano-probes to scan cellular or tissue samples, enabling the acquisition of comprehensive structural, quantitative, and spatial distribution information of numerous endogenous metabolites, exogenous drugs, and other molecules, irrespective of their prior characterization. This approach enables meticulous high-resolution spatial profiling and precise localization of metabolite distributions within tissues, thereby playing a critical role in elucidating the synthesis, accumulation, and regulatory mechanisms of metabolites (*Taylor, Lukowski & Anderton, 2021*; *Alexandrov, 2020*; *Alexandrov, 2023*; *Saunders et al., 2023*; *Chen et al., 2023a*).

MSI is a sensitive and efficient molecular imaging technique utilized for metabolite detection. It involves scanning biological tissue slice samples point by point using mass spectrometry and, in combination with specialized image processing software, directly analyzing a wealth of information regarding the composition, relative abundance, and spatial distribution of metabolites (Fig. 5). This methodology is widely employed in diverse areas including tumor diagnosis, identification of tumor biomarkers, and the exploration of drug distribution and mechanisms (*Kumar, 2023b*; *Kret et al., 2023*; *Norris & Caprioli, 2013*). MSI technology bypasses the need for preprocessing steps such as metabolite extraction, isotope labeling, or sample staining, enabling efficient imaging analysis of multiple substances within the sample and the visualization of metabolite spatiotemporal distribution. Depending on the ion source, MSI encompasses matrix-assisted laser desorption ionization-mass spectrometry imaging (MALDI-MSI), desorption electrospray ionization-mass spectrometry imaging (DESI-MSI), secondary ion-mass spectrometry imaging (SIMS-MSI), and laser ablation electrospray ionization-mass spectrometry imaging (LA-ESI-MSI), and so on. Currently, three primary spatial metabolomics technologies rely on MSI: MALDI-MSI, ambient flow-assisted desorption electrospray ionization-mass spectrometry imaging (AFADESI-MSI), and DESI-MSI (*Morato & Cooks, 2023*; *Watrous & Dorrestein, 2011*; *Schwamborn & Caprioli, 2010*).

In MALDI-MSI application for spatial metabolomics research, samples are coated or co-crystallized with light-absorbing matrices and then exposed to pulses from UV or IR lasers (*Norris & Caprioli, 2013*). The matrix absorbs the radiation, transferring energy to the sample and aiding ionization (Fig. 5). However, MALDI-MSI has limitations. It exhibits notable chemical noise at low mass ranges (<300 m/z) derived from matrix components, potentially hindering the ionization of critical small molecules and requiring samples to be positioned on conductive surfaces. Moreover, the process of laser desorption/ionization can cause sample damage. Notably, MALDI-MSI primarily detects metabolites such as proteins, peptides, and lipids. DESI-MSI technology operates on the principle of desorption electrospray ionization (Fig. 5). In contrast to MALDI-MSI, DESI-MSI necessitates less extensive sample preprocessing, induces minimal tissue destructiveness. These attributes enable DESI-MSI to bridge gaps in applications where other MSI methods fall short (*Morato & Cooks, 2023*). Currently, DESI-MSI is predominantly employed in the domain of medical research. Notably, studies have showcased DESI-MSI's capability

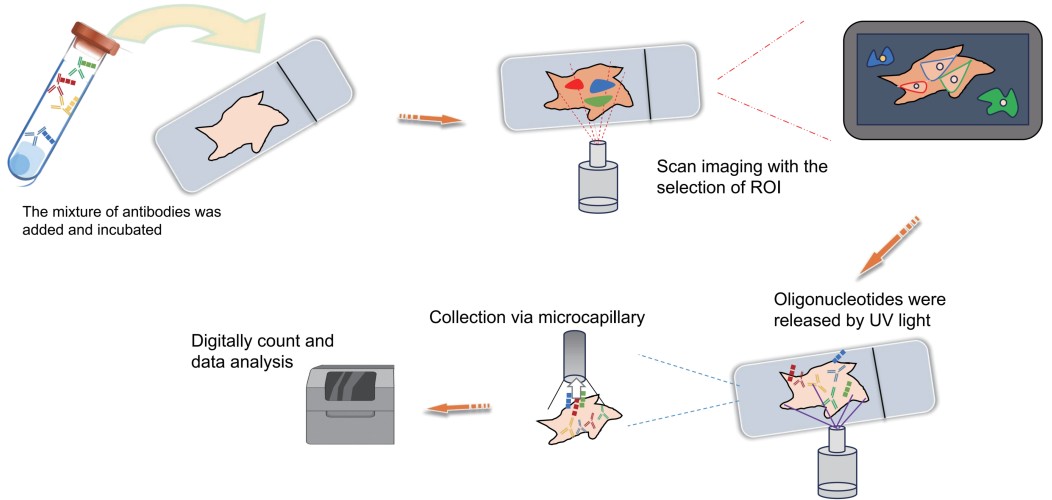

**Figure 5  Schematic diagrams of MADLI and DESI and AFADESI.** Sample preparation can involve both frozen or formalin-fixed paraffin-embedded (FFPE) specimens. In the processing phase, matrix-assisted laser desorption ionization (MALDI) primarily entails matrix-assisted laser desorption ionization for matrix-assisted laser desorption ionization (MALDI) for sample matrix deposition. On the other hand, desorption electrospray ionization (DESI) and air flow-assisted desorption electrospray ionization (AFADESI) primarily involve desorption electrospray ionization for ionization. AFADESI benefits from air flow-assisted transfer tubing to enhance ionization efficiency. Ultimately, all three techniques converge towards comprehensive mass spectrometry imaging analysis. Created with Figdraw, https://www.figdraw.com.

to directly extract comprehensive structural, quantitative, and spatial distribution data of both known and unknown endogenous metabolites, exogenous drug metabolites, and various other molecules from biological tissues (*Morato & Cooks, 2023*; *Kumar, 2023a*). AFADESI-MSI, an extension of DESI-MSI technology, effectively overcomes the bottleneck of limited metabolite identification in DESI-MSI (Fig. 5). It can map the spatial distribution of over 1,000 metabolites in tissue samples, thereby facilitating research into molecular mechanisms (*He et al., 2018*). In medical and clinical research, AFADESI-MSI has found extensive application in the study of disease molecular mechanisms, reproductive development, tumor metabolism and tumor immunity, tumor molecular pathological diagnosis, biomarker screening, pharmacology, and toxicology of drugs (*Parrot et al., 2018*; *He et al., 2015*).

In general, MALDI-MSI, DESI-MSI, and AFADESI-MSI technologies are utilized in spatial metabolomics research on large tissue samples. AFADESI-MSI is capable of detecting small molecule metabolites below 1,000 Da, including diverse categories such as choline, polyamines, amino acids, carnitines, nucleosides, nucleotides, organic acids, carbohydrates, cholesterol, bile acids, and lipids. DESI-MSI, although also capable of detecting small molecule metabolites below 1,000 Da, exhibits a preference for lipid metabolites. MALDI-MSI is proficient in identifying proteins, peptides, and lipid metabolites but has notable limitations in the detection of small molecule metabolites. With deeper research, the metabolic heterogeneity of cells and tissue samples has been well recognized. However,

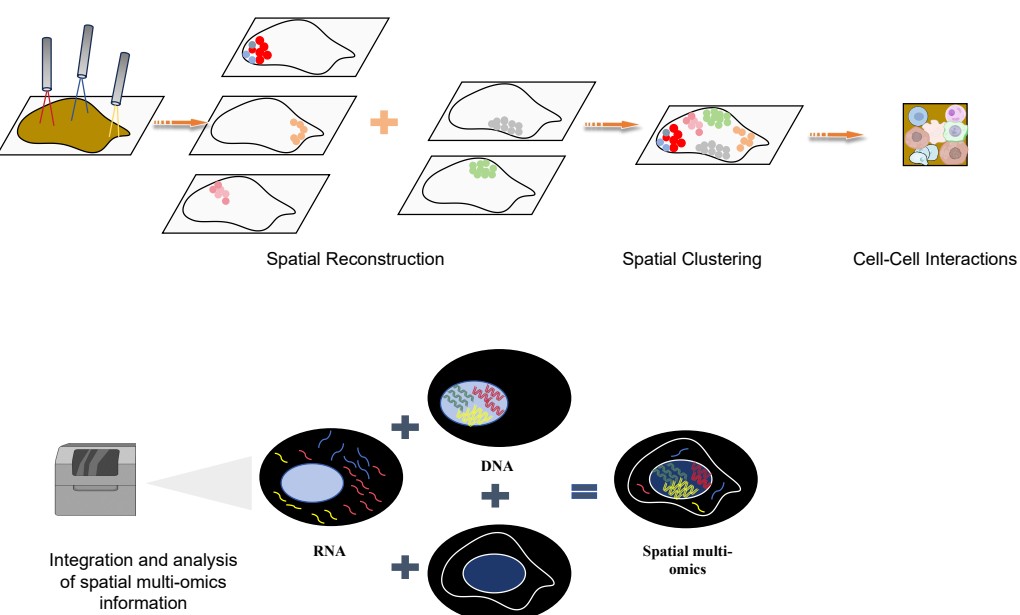

Spatial Reconstruction        Spatial Clustering        Cell-Cell Interactions

**Figure 6** **The flow chart of space group to learn more explanatory as a flowchart of spatial multiomics.** The spatial multi-omics technology was used to analyze the tissues at the single-cell resolution, and spatial clustering was performed through spatial reconstruction to analyze the interaction between cells. Finally, the collected spatial multi-omics information was integrated and analyzed. Created with Figdraw, https://www.figdraw.com.

the development of metabolomics technology has been relatively lagging due to the unsuitability of amplification and labeling strategies for metabolite molecules. Nonetheless, the combined utilization of various omics technologies has been integrated into research focusing on disease molecular mechanisms, tumor metabolism, tumor immunity, tumor molecular pathological diagnosis, and more (*Ravi et al., 2022*; *Lee et al., 2023*).

## Spatial multi-omics analysis of gastrointestinal tumor initiation and progression

Gastrointestinal cancer encompasses tumor formations in the digestive tract and digestive glands, with a higher incidence of cancer in the digestive tract and digestive glands (pancreas, liver, and gallbladder) compared to other parts of the human body (*Privitera et al., 2023*; *Ben-Aharon et al., 2023*). The number of deaths due to cancerous lesions in these areas is also relatively high. The development of multi-omics technologies allows us to delve into the secrets of cellular heterogeneity, the microenvironment of cells, and cellular communication during the process of tumor initiation and progression from various perspectives such as the genome, transcriptome, proteome, epigenetics, and metabolome (*Zhou et al., 2023*; *Zeng et al., 2023*). Spatial multi-omics techniques allow for more accurate 3D spatial information to be resolved at single-cell resolution, spatial clustering, and cell–cell interactions, while spatial multi-omics techniques have the advantage of having more diverse, three-dimensional sample information (Fig. 6). Here is a table comparing the popular spatial multi-omics technologies in these years

**Table 1 Examples of spatial multi-omics techniques.**

|  | DBiT-seq | Spatial CITE-seq | ATAC&RNA-seq | CUT&Tag-RNA-seq | GeoMx DSP |
|---|---|---|---|---|---|
| Tissue | Frozen + FFPE | FFPE + Frozen | FFPE + Frozen | FFPE + Frozen | FFPE + Frozen |
| Technology strategy | NGS + Antibody | NGS + Antibody | NGS | NGS + Antibody | NGS + Antibody |
| Output style | mRNA + Protein | RNA + Protein | Chromatin accessibility + RNA | Protein + RNA | RNA + Protein |
| Resolution | 10 μm | 20 μm | Single cell level | Single cell level | 1 μm |
| Sensitivity | High | Very High | Very High | High | Very High |
| Capture Efficiency | Very High | Very High | Very High | High | Very High |
| Cost | Low/medium | High | Low/medium | Low/medium | High |

(Table 1). Recent strides in spatial multi-omics technology have increasingly facilitated the exploration of tumor initiation and progression within a complex three-dimensional space (*Bressan, Battistoni & Hannon, 2023*). These advancements indicate an impending era of comprehensive comprehension regarding the emergence and evolution of tumor cells.

Compared with traditional multi-omics technology, spatial multi-omics technology can gain more special perspectives in gastrointestinal tumors: spatial resolution, cellular heterogeneity, multilevel analysis, and providing more accurate information for the construction of tumor disease models for researchers. Spatial resolution, spatial multi-omics technology can provide spatial resolution at the single-cell level, helping researchers to locate the exact position of cells in tissues and to observe the interactions between cells and neighboring cells or tissues, which provides strong evidence for understanding the complex tumor microenvironment (*Walsh & Quail, 2023*). Cellular heterogeneity, gastrointestinal tumors have a high degree of cellular heterogeneity and complexity, spatial multi-omics technology can identify the distribution and functional status of different cell populations in tumor tissues, which can help to monitor and characterize tumor development. Spatial multi-omics technology can analyze more detailed molecular spatial information at the molecular level of genome, proteome, metabolome, *etc.*, and the integrated information is more accurate than before. In summary, spatial multi-omics technology provides new perspectives and methods for the study of gastrointestinal tumors, which will advance the research development of gastrointestinal tumors. These features will provide insights into the cytomolecular diagnosis of gastrointestinal tumors, and as a basis for providing more refined means of treatment monitoring for gastrointestinal tumor patients, providing a basis for developing targeted therapy and individual precision therapy, and more accurate and refined scientific research will also provide a basis for promoting the progression of the disease and predicting the accuracy of treatment effects (*Kiessling & Kuppe, 2024*; *Xu et al., 2024*). By localizing and analyzing different tumor regions through this technology, researchers can more accurately and intuitively observe the impact of tumor heterogeneity on the effect of immunotherapy, which is crucial for predicting the response to treatment and developing personalized treatment strategies (*Song et al., 2023*). In recent years, studies on the application of spatial multi-omics technology in the prognosis of gastrointestinal tumor progression and other aspects have also been

demonstrated, which showed that spatial multi-omics technology revealed gastrointestinal tumor-specific metabolic remodeling and interactions and tracked molecular metabolites, genes, and lipids at metabolism-transcriptional level, which revealed the interaction between metabolic heterogeneity and cellular metabolism in tumors (*Sun et al., 2023*). Tumor drug resistance is closely related to immunity, and researchers have used this technology to track and validate the cellular interactions affected by drug resistance and provide insights into individual cancer treatment (*Che et al., 2024*). Another study showed that sequencing of colorectal cancer tissues to analyze changes in the macrophage immune microenvironment using DSP spatial genomics technology using imaging mass spectrometry flow (IMC) revealed significant changes in the relative abundance of specific macrophage populations (CD68, CD163, HLA-DR, CD204) from normal colorectal tissues to cancerous tissues (*Roelands et al., 2023*). These findings, using spatial multi-omics techniques, will help to identify novel therapeutic modalities and provide a reliable basis for the diagnosis of patient treatment, as well as new methods for monitoring tumor progression and prognosis.

Therefore, this article offers a detailed review of the application and progress of spatial multi-omics technology in three types of gastrointestinal cancers: esophageal cancer, gastric cancer, and colorectal cancer.

## Esophageal cancer

Esophageal cancer is one of the malignant tumors in the world with strong metastasis and high fatality rates. It is projected to yield approximately 57,000 new cases and 880,000 deaths by 2040 (*Morgan et al., 2022*). At present, esophagectomy stands as the primary therapeutic approach. Despite recent advancements in neoadjuvant chemotherapy, immunotherapy, and their combined application, the survival prospects for individuals afflicted with esophageal cancer persist at a disconcertingly low level (*Shah et al., 2023*). Histopathologically, esophageal cancer manifests in two principal forms: esophageal squamous cell carcinoma (ESCC) and esophageal adenocarcinoma (EAC) (*Smyth et al., 2017*). ESCC globally comprises approximately 90% of all esophageal cancer cases. Considered potential precursors to ESCC, esophageal squamous precursor lesions (ESPL) play a critical role in comprehending the onset and progression of ESCC and in devising innovative therapeutic interventions (*Wang et al., 2005*). However, the scarcity of research reports examining the pathogenesis from ESPL to ESCC can be attributed to the substantial coexistence of normal tissue within ESPL (*Liu et al., 2023*). While single-cell multi-omics sequencing can address the scarcity of ESPL cell samples, the complex sample preparation process can compromise the spatial information of heterogeneous tissue samples (*Bressan, Battistoni & Hannon, 2023*; *Moses & Pachter, 2022*; *Moffitt, Lundberg & Heyn, 2022*; *Seferbekova et al., 2023*; *Tian, Chen & Macosko, 2023*; *Vandereyken et al., 2023*; *Velten & Stegle, 2023*). Conversely, the development of spatial transcriptomics technology enables the sequencing of the transcriptome of ESPL samples containing a substantial amount of normal tissue cells in three-dimensional space at an approximate single-cell resolution, facilitating an in-depth exploration of the interplay between ESPL cell functionality, phenotype, and the microenvironment (*Bressan, Battistoni & Hannon,*

*2023*; *Moses & Pachter, 2022*; *Liu et al., 2023*; *Moffitt, Lundberg & Heyn, 2022*; *Seferbekova et al., 2023*; *Tian, Chen & Macosko, 2023*; *Vandereyken et al., 2023*; *Velten & Stegle, 2023*). *Liu et al. (2023)* employed nanostring whole-transcriptome analysis technology to profile the transcriptomes of 11 ROIs from normal esophagus (NE), 12 ROIs corresponding to low-grade intraepithelial neoplasia (LGIH), 12 ROIs associated with high-grade intraepithelial neoplasia (HGIH), and seven ROIs linked to ESCC. These ROIs were derived from 5 µm thick paraffin sections obtained from six LGIH patients, six HGIH patients, and seven ESCC patients. The investigation revealed an upregulation of TAGLN2 expression with the progression of ESCC, while CRNN exhibited an opposing trend, thereby unveiling novel biomarkers capable of distinguishing NE, ESPL, and ESCC. Nonetheless, nanostring whole-transcriptome analysis technology is subject to inherent limitations resembling those of bulk RNA-seq targeted at specific regions, restricting its capacity for achieving single-cell resolution. Moreover, the accuracy of RNA expression detection is contingent upon the characteristics of the employed nucleic acid probes (*Liu et al., 2023*). Compared to nanostring transcriptome analysis technology, the 10x Genomics Visium spatial transcriptomic sequencing technology offers a comprehensive depiction of spatial RNA transcription information in tumor samples (*Bressan, Battistoni & Hannon, 2023*; *Moses & Pachter, 2022*; *Moffitt, Lundberg & Heyn, 2022*; *Seferbekova et al., 2023*; *Tian, Chen & Macosko, 2023*; *Vandereyken et al., 2023*; *Velten & Stegle, 2023*). *Chen et al. (2023b)* performed single-cell RNA sequencing and spatial transcriptomic sequencing on a collective of 79 samples, encompassing NE, HGIH, and ESCC from 29 ESCC patients. Their findings highlighted the suppression of the epithelial cell transcription factor KLF4 with ESCC progression, leading to a gradual decline in ANXA1 expression (*Liu et al., 2022c*). ANXA1 functions as an FPR2 ligand, maintaining fibroblast homeostasis, and its absence contributes to the uncontrolled conversion of normal fibroblasts into cancer-associated fibroblasts (CAFs) (*Chen et al., 2023b*). Simultaneously, the combined utilization of single-cell transcriptomic sequencing and spatial transcriptomics furnishes spatial characteristic insights into distinct cell subpopulations within the ESCC tumor microenvironment (*Guo et al., 2022*). Consequently, this dual approach not only mitigates the inherent limitations of spatial information loss during sample preparation in single-cell transcriptomic sequencing but also addresses the spatial transcriptomics' constraints concerning single-cell resolution (*Bressan, Battistoni & Hannon, 2023*; *Moses & Pachter, 2022*; *Moffitt, Lundberg & Heyn, 2022*; *Seferbekova et al., 2023*; *Tian, Chen & Macosko, 2023*; *Vandereyken et al., 2023*; *Velten & Stegle, 2023*).

Tumor metabolic reprogramming significantly influences tumor initiation and progression, displaying notable heterogeneity and variability (*Li et al., 2023b*; *Finley, 2023*). Spatial metabolomics technology spatially characterizes metabolites and enzymes involved in tumor metabolic reprogramming, enhancing our understanding of the complex process. This approach aids in identifying potential metabolic vulnerabilities, thereby facilitating the development of targeted therapies reliant on these vulnerabilities (*Yuan et al., 2021*; *Vicari et al., 2023*). Chenlong Sun and colleagues employed their developed AFADESI-MSI technology, enabling the mapping of diverse functional metabolites across various metabolic pathways (*He et al., 2018*). Their spatial metabolomic investigation encompassed

256 cases of cancer alongside corresponding normal tissues. The study identified notable modifications in the proline biosynthesis, glutamine metabolism, uridine metabolism, histidine metabolism, fatty acid biosynthesis, and polyamine biosynthesis pathways specific to ESCC. Furthermore, the aberrant expression of six metabolic enzymes, namely PYCR2, GLS, UPase1, HDC, FASN, and ODC, was concurrently observed throughout the ESCC carcinogenesis process (*Sun et al., 2019*).

The combined application of spatial metabolomics and scRNA-seq technologies allows for the precise detection of mRNA transcripts and low-molecular-weight metabolites within tissue sections. This integrated approach not only offers mRNA transcriptional insights but also provides valuable spatial metabolic information, thereby enhancing our understanding of the intricate relationship between tissue metabolism and homeostasis (*Vicari et al., 2023*). However, current research on the concurrent use of spatial multi-omics techniques in esophageal cancer remains limited. Techniques such as spatial ATAC-seq and spatial-RNA-seq joint analysis, spatial proteomics combined with spatial transcriptomics and spatial metabolomics analysis, as well as spatial transcriptomics combined with spatial epigenomics analysis, have not been reported in esophageal cancer research (Table 2). Certainly, the application of these emergent spatial multi-omics technologies in esophageal cancer will significantly advance our understanding of esophageal cancer tumor biology and foster the development of effective therapeutic strategies.

## Gastric cancer

Gastric cancer, as the third most fatal cancer originating in the stomach, predominantly manifests as gastric adenocarcinomas, accounting for 90% of cases (*Hirata et al., 2023*). These malignancies emerge from the glandular epithelium of the gastric mucosa. Histologically classified into intestinal and diffuse types according to the Lauren system, intestinal-type gastric adenocarcinoma prevails over the diffuse type (*Hirata et al., 2023*; *Thrift, Wenker & El-Serag, 2023*). Typically, the former exhibits a well-defined stepwise progression often triggered by chronic inflammatory mucosal damage, including instances induced by Helicobacter pylori infection. On the other hand, the origin of diffuse-type gastric cancer remains ambiguous, potentially linked to gene mutations affecting pathways associated with cell-extracellular matrix interactions (*Hirata et al., 2023*; *Thrift, Wenker & El-Serag, 2023*). Tumor heterogeneity poses a significant impediment in the progression of gastric cancer therapy, such as the genetic distinctions between primary and metastatic gastric cancer that impede the advancement of precision oncology (*Hirata et al., 2023*; *Röcken et al., 2021*). Clinically, the diagnosis and treatment selection for gastric cancer are typically conducted by assessing endoscopic biopsies obtained from the luminal part of the primary tumor ('superficial mucosa') (*Shiotani, Cen & Graham, 2013*). Nonetheless, the presence of spatial intratumoral heterogeneity within the gastric cancer tumor may substantially influence the outcomes of these biopsies. Therefore, *Sundar et al. (2021)* undertook the characterization of 64 specific subregions, including the superficial and deep sections of the primary tumor, as well as areas of lymph node metastasis, employing nanostring transcriptomic analysis technology (the panel comprising 770 genes). Their findings illuminated notable heterogeneity in mRNA expression among these subregions,

**Table 2** The application of spatial profiling technologies in gastrointestinal cancer.

| Tumors | Spatial profiling technologies | Samples | Year |
|---|---|---|---|
| Esophageal squamous cell carcinoma | 10x Genomics Visium spatial transcriptomic sequencing, in addition to scRNA-seq | Three patients who were diagnosed with pathologically confirmed ESCC | 2022 (*Guo et al., 2022*) |
| Esophageal squamous cell carcinoma | AFADESI-MSI | A total of 256 pairs of matched human ESCC tissue samples, including cancer tissues, adjacent noncancerous tissues | 2019 (*Sun et al., 2019*) |
| Esophageal squamous-cell carcinoma | 10x Genomics Visium spatial transcriptomic sequencing, in addition to scRNA-seq | A total of 29 samples from esophageal squamous cell carcinoma patients, including normal esophagus, high-grade intraepithelial neoplasia, and esophageal squamous cell carcinoma | 2023 (*Chen et al., 2023b; Liu et al., 2022c*) |
| Esophageal cancer | Nano-string transcriptomic analysis (GeoMx WTA panel) | Six patients with low-grade intraepithelial neoplasia, six patients with high-grade intraepithelial neoplasia, and seven patients with esophageal squamous cell carcinoma | 2023 (*Liu et al., 2023*) |
| Gastric cancer | Nano-string transcriptomic analysis (the panel of 770 genes), in addition to the next-generation sequencing (225 targeted genes), DNA copy number profiles by multiplex ligation-dependent probe amplification. | 64 gastric cancers resection samples, including tumor superficial, primary tumor deep and lymph node metastasis subregions | 2020 (*Sundar et al., 2021*) |
| Gastric cancer | Nano-string transcriptomic analysis (the panel of 1,812 genes), in addition to scRNA-seq and bulk RNA-seq | scRNA-seq (about 200,000 cells): 48 surgical resection and biopsy samples across 31 patients with gastric cancer, ranging from clinical stages and histologic subtypes. Nanostring: 13 samples including 10 tumor and three normal, 156 regions of interest | 2022 (*Kumar et al., 2022*) |
| Gastric cancer | Microarray-based spatial transcriptomics (ST), in addition to the mass spectrometry imaging-based spatial metabolomics (SM) and lipidomics (SL) | Postoperative cancer tissue from seven male patients diagnosed with gastric adenocarcinoma. SM: Two sets of tissue sections ($n = 7$). SL: Two sets of adjacent sections ($n = 7$) ST: Tissue sections which adjacent to the ones used for SM and SL ($n = 4$) | 2023 (*Sun et al., 2023*) |
| Gastric cancer | SM | Primary resected gastric cancer samples were obtained from 362 patients who underwent gastrectomy between 1995 and 2005 at the Surgery Department at the Technical University Munich (tissue microarrays were analyzed in three tissue cores from each patient). | 2022 (*Wang et al., 2022*) |
| Gastric cancer | Nanostring transcriptomic analysis (the panel of 1,850 genes) | Samples from nine patients with gastric cancer | 2023 (*Park et al., 2023*) |
| Gastric cancer | ST | Four gastric cancer primary tumors | 2023 (*Jang et al., 2023*) |

| Tumors | Spatial profiling technologies | Samples | Year |
|---|---|---|---|
| Gastric cancer | Nanostring transcriptomic analysis (the panel of 31 genes) | 130 tissue microarray cores from 49 patients | 2023 (*Choi et al., 2023*) |
| Colorectal cancer | ST, in addition to scRNA-seq and bulk RNA-seq | 89 samples from 20 patients underwent scRNA-seq and eight samples from four patients were sequenced by ST | 2022 (*Wu et al., 2022b*) |
| Colorectal cancer | GeoMx transcriptomics, in addition to scRNA-seq and CyCIF. | 93 FFPE CRC human specimens | 2023 (*Lin et al., 2023*) |
| Colorectal cancer | ST, in addition to scRNA-seq and GeoMx digital spatial profiling (77 proteins) | ST to a specimen of CRC | 2022 (*Galeano Niño et al., 2022*) |
| Oral squamous cell carcinoma | ST, in addition to scRNA-seq and GeoMx digital spatial profiling (77 proteins) | ST to a specimen of OSCC | 2022 (*Galeano Niño et al., 2022*) |
| Colorectal cancer | Matrix assisted laser desorption ionization (MALDI) image-guided proteomics | Eight fresh human colorectal carcinoma liver metastases | 2014 (*Turtoi et al., 2014*) |
| Colorectal cancer | Digital Spatial Profiling (DSP), 84 genes at the transcriptional level and 40 at the protein level in all ROIs | Tumor specimens from four patients | 2021 (*Wang et al., 2021b*) |
| Colorectal cancer | Digital Spatial Profiling (DSP), 1,825 CTA genes. | 5 $\mu$m thick FFPE sections of eight different patients | 2023 (*Roelands et al., 2023*) |
| Colorectal cancer | Spatial lipidomics by MALDI-MSI | A colorectal cancer tissue microarray (TMA, $n = 30$) | 2021 (*Denti et al., 2021*) |
| Colorectal cancer | Digital Spatial Profiling (DSP), 40 at the protein level | 36 resected colorectal tumor specimens | 2023 (*Levy et al., 2023*) |
| Colorectal cancer | Digital Spatial Profiling (DSP), about 1,400 genes at the transcriptional level | Paired epithelial and non-epithelial regions from three patients | 2021 (*Pelka et al., 2021*) |
| Colorectal cancer | ST, in addition to scRNA-seq and metabolic profiling | Tumor specimens from six patients | 2023 (*Fleischer et al., 2023*) |
| Colorectal cancer | ST (10x Genomics Visium), in addition to scRNA-seq | Tumor specimens from four patients | 2022 (*Qi et al., 2022*) |
| Colorectal cancer | ST (10x Genomics Visium), in addition to scRNA-seq | Tumor specimens from six patients | 2023 (*Wang et al., 2023*) |

implying that regional lymph node metastases are likely to originate from the deeper segments of the primary tumor (*Sundar et al., 2021*). Furthermore, the integration of nanostring transcriptomic analysis technology with scRNA-seq by *Kumar et al. (2022)* yielded an extensive database featuring over 2 million single-cell sequencing profiles for gastric cancer. This resource provides a unique opportunity to comprehend the cellular subtypes within gastric cancer tumors, the intricate composition of the gastric cancer tumor microenvironment based on subtypes, and the intricate cellular interactions within the gastric tumors. Consequently, the synergistic deployment of single-cell omics and spatial transcriptomics emerges as an effective and promising research strategy in comprehending the dynamics of occurrence and development of gastric cancer.

In addition to the synergy between spatial transcriptomics and single-cell multi-omics, a combination of various spatial multi-omics technologies can also be used to facilitate the elucidation of tumor metabolic reprogramming. *Sun et al. (2023)* conducted spatial metabolomics, spatial lipidomics, and spatial transcriptomics analyses on adjacent frozen sections of tumor tissue from patients with gastric adenocarcinoma. Their findings

underscore the value of integrating spatial metabolomics, spatial lipidomics, and spatial transcriptomics for the investigation of the markedly heterogeneous tumor tissues in gastric cancer. This integrative methodology yields a comprehensive depiction of the tumor's metabolic landscape, emphasizing the intricate interplay between metabolites and lipids within the metabolic network (*Sun et al., 2023*; *Wang et al., 2022*). Moreover, it enables the visualization of gene regulatory networks linked to metabolite and lipid expression within the framework of tumor metabolic reprogramming. According to existing research, nanostring transcriptomic analysis has shown greater prevalence in human gastric cancer samples compared to other spatial omics technologies, as indicated in Table 2. Anticipated increases in the publication of spatial multi-omics research findings on human gastric cancer samples are expected. These resources promise to substantially enhance our comprehension of the development and progression of human gastric cancer, potentially leading to the discovery of novel treatment modalities.

## Colorectal cancer

Colorectal cancer (CRC), known as bowel, colon, or rectal cancer, originates in the colon or rectum (components of the large intestine) (*Spaander et al., 2023*; *Bando, Ohtsu & Yoshino, 2023*). The application of spatial multi-omics technology in colorectal cancer research is rapidly deepening our understanding of this disease.

Despite our comprehensive understanding of the genomic determinants of colorectal cancer, the impact of the spatial arrangement of the tumor microenvironment and intratumoral heterogeneity on the initiation of colorectal cancer remains incomplete. Spatial tumor mapping collects detailed information about cellular molecules and morphology in a 3D context, integrating it with tumor genetics (*Rozenblatt-Rosen et al., 2020*; *Lin et al., 2023*). This process sheds light on the tumor microenvironment and intratumoral heterogeneity. Researchers have utilized the MALDI-MSI to identify consistent, distinct protein heterogeneity patterns in human liver metastases of colorectal cancer (*Turtoi et al., 2014*). Moreover, the amalgamation of spatial multi-omics technology with scRNA-seq techniques enables the characterization of heterogeneity in cells, molecules, and morphology within the colorectal cancer tumor microenvironment. *Lin et al. (2023)* employed high-plex cyclic immunofluorescence (CyCIF) technology in conjunction with scRNA-seq and GeoMx transcriptomics to characterize 93 FFPE CRC human specimens, constructing a spatial tumor map. Their findings demonstrated the stratification of molecular states (protein markers) and tissue morphologies (histotypes) within colorectal cancer tissue, ranging spatially from cell sizes to several millimeters. Additionally, cell populations that are typically studied in a 2D manner at the local level are frequently organized into large, interconnected 3D structures. Moreover, the combined application of scRNA-seq and spatial transcriptomics techniques in characterizing colorectal cancer liver metastases unveiled a significant temporal and spatial remodeling of the immune microenvironment during the metastatic progression (*Wu et al., 2022b*). Noteworthy observations included an enrichment of immunosuppressive cells and the identification of highly metabolically active MRC1[+] CCL18[+] M2-like macrophages at the metastatic site. These macrophages exhibited a terminal differentiation state,

showcasing a robust metabolic phenotype and displaying sensitivity to neoadjuvant chemotherapy. These findings enhance our understanding of how immune cells within the tumor microenvironment spatially orchestrate the progression of colorectal cancer liver metastasis. Moreover, the presence of diverse microorganisms in the heterogeneous tumor microenvironment is pertinent to tumor initiation and development, amenable to investigation through scRNA technology and spatial transcriptomics. For example, the combined utilization of scRNA technology and spatial transcriptomics has unveiled the diversity of tumor microbiota in human oral squamous cell carcinoma and colorectal cancer (*Galeano Niño et al., 2022*). Despite this, the current utilization of spatial multi-omics technology in the context of colorectal cancer remains constrained (Table 2). We believe that the increasing application of multi-omics technology in colorectal cancer will rapidly deepen our understanding and treatment of this disease.

The following is a summary of specific molecular markers for the diagnosis or treatment of GI tumors revealed by multi-omics techniques (Table 3).

## CONCLUSIONS

The complexity of human diseases unfolds within intricate three-dimensional environments. The advent of spatial multi-omics technology has expedited our comprehension of disease onset and progression, fostering novel avenues for treatment development. Nonetheless, challenges persist in spatial multi-omics technology, encompassing limited single-cell resolution, high costs, and considerable application thresholds. While the combined usage of spatial multi-omics technology, single-cell multi-omics technology, and bulk multi-omics technology can address some shortcomings, such as challenges in single-cell resolution and low molecular capture efficiency, intricate bioinformatics analysis and significant expenses remain additional barriers. In the foreseeable future, the widespread commercialization of spatial multi-omics technology, single-cell multi-omics technology, and bulk multi-omics technology will significantly propel the advancement and utilization of spatial multi-omics technology, enriching our grasp of disease dynamics. The combination of spatial omics technologies and traditional technologies in gastrointestinal tumors will take advantage of the spatial resolution of spatial omics technologies and the convenience of traditional technologies. Bulk RNA seq and standard proteomics have higher efficiency and lower cost when processing a large number of samples, which is suitable for large number of samples to be screened and analyzed for research, but usually mix the analysis of whole tissue samples, which may result in the loss of information about the heterogeneity of different regions within a tumor (*Kuksin et al., 2021*; *Laurinavicius et al., 2016*). The *in situ* analysis provided by spatial omics technologies provides reliable information to reveal the original space of the host and cell-molecule interactions (*Galeano Niño et al., 2022*). Bulk RNA seq and standard proteomics have a wider coverage, while spatial omics technologies relies on specific high-end equipment to obtain more accurate spatial information in a specific region, which will greatly reduce its coverage width, convenience and popularity. This will greatly reduce its coverage width, convenience and popularity, we can utilize the broad coverage of classical omics techniques

**Table 3 Specific molecular markers of interest to diagnose or treat specifically GI cancer revealed by the modern multi-omics techiques.**

| Multi-omics techiques | Technical field | Specific molecular markers | Research significance |
|---|---|---|---|
| AFADESI-MSI + MALDI-MSI | Spatial metabolomics + Spatial lipidomics | Arginine, Proline, Glutamate, Glutamine, fatty acid:Palmitic Acid(FA-16:0), Arachidonic Acid (FA-20:4) | Lipid and metabolites of molecular diagnosis and treatment to provide value (*Sun et al., 2023*) |
| | | Phospholipids: Phosphatidylcholine 32:0, Phosphatidylethanolamine 34:0, Phosphatidylcholine 34:1, Phosphatidylethanolamine 36:1, Phosphatidylinositol 34:1, Phosphatidylinositol 36:1 | |
| | | glucose metabolism:Lactic Acid, Succinic Acid, Malic Acid | |
| | Spatial transcriptomics | Glutamate + Glutamine:GLUL, GLS, Arginine + Prolin:ASS1, ALDH18A1 , PYCR, OAT, AGMAT, ODC1, SRM, SMS | Key genes that regulate metabolic pathways (*Sun et al., 2023*) |
| | | fatty acid:SCD, FADS, ELOVL | |
| | | Phospholipids:ETNK1, CHKA, PLD, LYPLA2 | |
| | | glucose metabolism:NDUFS6, NDUFA6, NDUFAB1, NDUFB4, NDUFB3, COX5A, COX7B, COX7A2, UQCR11, UQCR10, UQCRQ, ATP5MC3, ATP5F1E, ATP5PF | |
| DESI-MSI + Immunohistochemistry (IHC) + Quantitative reverse transcriptase-PCR (qRT-PCR) | Spatial metabolomics + Standard proteomics + Standard transcriptomics | ACLY | Esophageal adenocarcinoma of lipid metabolic pathway in cancer targets (*Abbassi-Ghadi et al., 2020*) |
| GeoMx DSP + Immunohistochemistry (IHC) + Immunofluorescence (IF) + Western Blot | Spatial transcriptomics + Standard proteomics | TAGLN2, CRNN | Potential predictors of the risk of ESCC (*Liu et al., 2023*) |
| DSP + Immunohistochemistry (IHC) + mIF + Western Blot + Gene ChIP Assay | Spatial transcriptomics + Spatial Proteomics + Standard proteomics | GLI, PD-L1, mTOR, Arg1, CD66b, VISTA, IDO1 | In gastric cancer, immune escape and immune potential therapeutic value of protein (*Koh et al., 2021*) |
| scRNA-seq + DSP + Immunohistochemistry (IHC) + Western Blot | Single-cell transcriptomics + Spatial transcriptomics + Standard proteomics | KLF2, INHBA, FAP, PLVAP, RGS5 | Potential therapeutic targets in gastric cancer (*Kumar et al., 2022*) |

| Multi-omics techiques | Technical field | Specific molecular markers | Research significance |
|---|---|---|---|
| scRNA-seq + Spatial transcriptomics + Immunohistochemistry (IHC) | Single-cell transcriptomics + Spatial transcriptomics + Standard proteomics | ALKBH1, AQP1, PECAM1 | In gastric cancer, the diagnostic tools, prognostic evaluation index, the potential of molecular drug targets and immunotherapy (*Chang et al., 2024*) |
| | | CD83, TNFRSF4, TNFSF14, VEGFR2, ADA, ARG1, HO-1 | Forecast and manage the potential adverse reactions of protein in the process of esophageal cancer treatment (*Zhang et al., 2023*) |
| Bulk RNA-seq + Whole-Exome Sequencing (WES) + mIF + Immunohistochemistry (IHC) | Genomics + Transcriptomics + Spatial proteomics | PD-L1, TMB, TNB, CD4 + T | Potential proteins with predictive response and prognostic value in esophageal cancer (*Zhang et al., 2023*) |
| scRNA-seq + mIHC + Western Blot | Single-cell transcriptomics + Spatial proteomics + Standard proteomics | TGFB1, HSPB1 | Potential molecular markers for gastric cancer chemotherapy resistance (*Che et al., 2024*) |
| DSP + IMC + scRNA-seq + IHC | Spatial genomics + Single-cell transcriptomics + Standard proteomics | MUC4, IFITM1, CD81, NOTCH3, PDGFRB, Thy1, Hsp47, CD47-SIRP $\alpha$ | Clinically relevant biomarkers and therapeutic targets for early development and progression (*Roelands et al., 2023*) |
| ISH + mIHC/mIF + scRNA-seq | Spatial proteomics + Single-cell transcriptomics | Stem Cell Index, LGR5, ANXA1 | Colon cancer chemotherapy response potential evaluation index:Stem Cell Index (*Vasquez et al., 2022*) |
| mIHC + scRNA-seq + Spatial transcriptomics | Spatial proteomics + Spatial transcriptomics + Single-cell transcriptomics | MRC1 + , CCL18 + M2-like macrophages, IL4I1, MIF | Potential therapeutic target molecules related to liver metastasis (*Wu et al., 2022b*) |
| scRNA-seq + Spatial transcriptomics + Immunofluorescence (IF) | Single-cell transcriptomics + Spatial transcriptomics + Spatial proteomics | FAP + , SPP1 + | Treatment of potential target cell interactions (*Qi et al., 2022*) |
| mIHC/mIF + Spatial transcriptomics + Western Blot + Immunoprecipitation (IP) + Quantitative reverse transcription polymerase chain reaction(qRT-PCR) | Spatial proteomics + Spatial transcriptomics + Standard proteomics + Standard transcriptomics | USP14, IDO1, TRIM21 | Immunotherapy and surveillance of immune escape (*Shi et al., 2022*) |
| scRNA-seq + mIHC + Immunohistochemistry (IHC) Quantitative real-time reverse transcription PCR (qRT-PCR) | Spatial proteomics + Single-cell transcriptomics + Standard transcriptomics | TCF-1, $\gamma\delta$ T-IELs | Immunotherapy and prognostic value (*Yakou et al., 2023*) |
| mIHC + RNA-seq | Classical omics techniques + Spatial proteomics | XBP1 | Value of potential therapeutic targets (*Zhao et al., 2021*) |

for coarse screening, and then combine with spatial omics technologies for more accurate analysis of microscopic information. bulk RNA seq and standard proteomics are suitable for rapid screening and identification of relevant gene expression patterns, while spatial omics technologies is more suitable for microscopic perspectives, for example, spatial omics technologies is more suitable for the analysis of spatial information. The development of classical omics techniques is relatively mature, while spatial omics technologies is a new and rapidly developing technology that requires a long period of application exploration, and the respective features of classical omics techniques and spatial omics technologies complement each other and are expected to play a more functional role. The respective features of classical omics techniques and spatial omics technologies are complementary to each other and are expected to fulfill more functions.

In the context of digestive tumor diseases, the application of spatial multi-omics technology is still limited, and the types of digestive tumor diseases involved are also restricted, with relatively fewer studies involving the application of various multi-omics technologies. The molecular mechanisms underlying the occurrence and development of digestive tumor diseases involve multiple aspects such as genomics, transcriptomics, epigenetics, and metabolomics. Therefore, efforts to comprehensively analyze the molecular mechanisms related to digestive tumor diseases from multiple dimensions will bring us closer to the truth of the occurrence and development of digestive tumor diseases. Animal models related to digestive tumor diseases are indispensable tools for studying the occurrence and development processes. Nevertheless, to date, there have been few studies on the application of spatial multi-omics technology in animal models of digestive tumor diseases. Ultimately, the extensive integration of spatial multi-omics technology in both clinical and animal model samples will significantly enrich our understanding of the molecular underpinnings of digestive tumor diseases. The more complex cellular complexity and heterogeneity in gastrointestinal tumors are involved in a more complex tumor microenvironment, nowadays immunotherapy is a hotspot for development, and a full understanding of the tumor microenvironment is essential for the development of immunotherapy, and the use of spatial multi-omics technology in the cell cell–cell heterogeneity and cell–cell interaction provides a new perspective, the tumor microenvironment involves the intercellular communication and interactions are Quite complex, and the fine positioning, spatial location information between cells and cells or tissues which is quite important, through the development of spatial multi-omics technology will also open up the micro three-dimensional world of spatial information acquisition and analysis, so that the location, behavior, function, and interactions between individual cells become clear. However, spatial multi-omics technology is an emerging field of technology, which requires special refined equipment and relatively high cost, and many of the technologies are still in the early stage, which need to be further optimized and standardized (*Bressan, Battistoni & Hannon, 2023*), and because it involves multi-level location information, the amount of three-dimensional information involved is huge, which still requires complex bioinformatics analysis tools to integrate and analyze. Although spatial multi-omics technology has also been applied in clinical research with remarkable results in recent years, there is still a lack of large-scale clinical studies to validate

the practical role in the diagnosis and treatment of gastrointestinal tumors in the face of the heterogeneity of the microenvironment in complex gastrointestinal tumors (*Wu et al., 2022a*). Despite the great potential of spatial multi-omics technology in gastrointestinal tumor research, further technological development and optimization are needed for better application in clinical practice. It is expected that these limitations and unresolved issues will be resolved as technology advances and costs decrease.

### Abbreviations

| | |
|---|---|
| NGS | Next-generation sequencing |
| HDST | High-Definition Spatial Transcriptomics |
| scRNA-seq | Single-cell RNA sequencing |
| DSP GeoMx | Digital Spatial Profiler |
| DBiT-seq | Deterministic Barcoding in Tissue for spatial omics sequencing |
| ISS | *In Situ* Sequencing |
| ISH | *In Situ* Hybridization |
| FISSEQ | Fluorescence *In Situ* Sequencing |
| smFISH | single-molecule fluorescence *in situ* hybridization |
| SeqFISH | Sequential Fluorescence *In Situ* Hybridization |
| MERFISH | Multiplexed Error-Robust Fluorescence *In Situ* Hybridization |
| MSI | Mass spectrometry imaging |
| MALDI-MSI | Matrix assisted laser desorption ionization-mass spectrometry imaging |
| ROIs | Regions of interest |
| MIBI | Multiplexed Ion Beam Imaging |
| MIBI-TOF | Multiplexed Ion Beam Imaging-Time of Flight |
| FFPE | Formalin-fixed paraffin-embedded |
| CODEX | Co-Detection by Indexing |
| Da | Daltons |
| DESI-MSI | Desorption electrospray ionization-mass spectrometry imaging |
| SIMS-MSI | Secondary ion-mass spectrometry imaging |
| LA-ESI-MSI | Laser ablation electrospray ionization-mass spectrometry imaging |
| ESCC | Esophageal squamous cell carcinoma |
| EAC | Esophageal adenocarcinoma |
| ESCC | Esophageal squamous cell carcinoma |
| ESPL | Esophageal squamous precursor lesions |
| NE | Normal esophagus |
| LGIH | Low-grade intraepithelial neoplasia |
| HGIH | High-grade intraepithelial neoplasia |
| CAFs | Cancer-associated fibroblasts |
| CRC | Colorectal cancer |
| AFADESI-MSI | Ambient Flow-Assisted Desorption Electrospray Ionization Mass Spectrometry Imaging |
| IHC | Immunohistochemistry |
| qRT-PCR | Quantitative reverse transcriptase-PCR |
| mIF | multiplex immunofluorescence |

| mIHC | Multiplex immunohistochemical |
| IMC | Imaging Mass Cytometry |
| IF | Immunofluorescence |
| IP | Immunoprecipitation |

## ACKNOWLEDGEMENTS

We thank Ms. Hongjiao Wang for checking the language of the manuscript.

### Funding

This work was supported by the Zhangjiakou City Key R&D Plan Project (Grant No. 2322088D and 2311038D), the Hebei Provincial Natural Science Foundation (H2022405033) and the Hebei Province Key R&D Plan Project (22377784D). The funders had no role in study design, data collection and analysis, decision to publish, or preparation of the manuscript.

### Grant Disclosures

The following grant information was disclosed by the authors:
Zhangjiakou City Key R&D Plan Project: 2322088D, 2311038D.
Hebei Provincial Natural Science Foundation: H2022405033.
Hebei Province Key R&D Plan Project: 22377784D.

### Competing Interests

The authors declare there are no competing interests.

### Author Contributions

- Weizheng Liang conceived and designed the experiments, performed the experiments, analyzed the data, prepared figures and/or tables, authored or reviewed drafts of the article, and approved the final draft.
- Zhenpeng Zhu conceived and designed the experiments, analyzed the data, prepared figures and/or tables, and approved the final draft.
- Dandan Xu performed the experiments, analyzed the data, authored or reviewed drafts of the article, and approved the final draft.
- Peng Wang performed the experiments, analyzed the data, authored or reviewed drafts of the article, and approved the final draft.
- Fei Guo performed the experiments, analyzed the data, authored or reviewed drafts of the article, and approved the final draft.
- Haoshan Xiao analyzed the data, prepared figures and/or tables, and approved the final draft.
- Chenyang Hou analyzed the data, prepared figures and/or tables, and approved the final draft.

- Jun Xue conceived and designed the experiments, analyzed the data, prepared figures and/or tables, authored or reviewed drafts of the article, and approved the final draft.
- Xuejun Zhi conceived and designed the experiments, analyzed the data, authored or reviewed drafts of the article, and approved the final draft.
- Rensen Ran analyzed the data, authored or reviewed drafts of the article, and approved the final draft.

## Data Availability

This is a literature review.

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
