# Peer review of "The burgeoning spatial multi-omics in human gastrointestinal cancers"

_PeerJ, doi:10.7717/peerj.17860_

## Round 0.1 · original submission · Major Revisions

1. The flow chart of this review should be provided.
2. Please make sure to explain all the acronymous the first time you encounter them within the text.
3. The manuscript contains numerous syntax, typographic and/or grammatical errors. It is thus strongly recommended that the revised manuscript be edited by a credible professional language editing service.

**Language Note:** The Academic Editor has identified that the English language must be improved. PeerJ can provide language editing services - please contact us at [email protected] for pricing (be sure to provide your manuscript number and title). Alternatively, you should make your own arrangements to improve the language quality and provide details in your response letter. – PeerJ Staff

Reviewer 1 ·

Basic reporting

This review is on the "burgeoning spatial multi-omics in human gastrointestinal
cancers".

The manuscript is a good start: Long overview on the spatial multi-omics techniques,
then citing some examples of the application of these techniques on gastrointestinal
cancers by some groups you know well and have demonstrated success there.

However, it would be far more interesting to read if the review has more focus and novelty:

1. Spatial multi-omics: Have an explanatory figure and a Table comparing the techniques,
otherwise be brief summarizing the spatial omics techniques and not unnecessary lengthy
as you are now. Point out good reviews or methods papers for each spatial omics technique
for the interested reader, far better then to write lengthy yourself.

2. The topic of your review was chosen by yourself to be "multi-omics in human gastrointestinal
cancers". The reader does not expect to see a collection of some chosen application of multi-omics
techniques from your own lab, and associated / regional / national groups: it is pretty clear, these
advanced techniques can now be done by any strong research group in the field in many countries.

Instead, you have to speak about the problems to identify key factors important in human gastrointestinal (GI) cancer:
what can one spatial omics method reveal regarding GI cancer which was not seen before?
How did this help in diagnosis or even in therapy? Could you monitor GI cancer patients?
Could you reveal new factors in GI cancer progression?

3. It would also be very nice to show where the modern spatial omics techniques complement
classical omics techniques (bulk RNAseq, standard proteomics) in giving insights on GI cancer.

4. To conclude speak about limitations and unresolved questions in GI cancer which may soon
become more clear using spatial omics even more advanced then now.

Experimental design

see above

Validity of the findings

see above

Additional comments

see above

·

Basic reporting

Authors of this work have attempted to contribute to the global literature library on the subject of multi-omics and cancer diagnosis and therapy. This review is highly beneficial and the authors have shown a professional understanding and grasp of the subject matter. However, some key revisions are suggested.

Experimental design

This study was campaigning for spatial multi-omics technology as an advantageous technology over the relatively more conventional single-cell multi-omics. Authors ought to commence the manuscript with a succinct review description of single-cell multi-omics with emphasis on its limitations and how significant those limitations are. There should be a section that thoroughly offers a discourse on this. This point was well established in the abstract and introduction aspects, however, a more detailed review in a dedicated section prior to the commencement of the review on spatial multi-omics technology will make things better.

Validity of the findings

In addition, authors should state and review (if available) studies that have compared both multi-omics technologies for the same type of cancer. This will further assist in validation of the point that the spatial route is superlative. A summary of comparisons can also be made into a figure that can function as the graphical abstract of the review.

---

## Round 0.2 · Minor Revisions

You paper will be accepted after the minor revision suggested by R1.

Reviewer 1 ·

Basic reporting

The revision nicely illustrates multi-omics techniques and how they are useful in cancer.
We even have Table 2 showing multi-omics technology applied to GI cancers, fine.

Open comment 1 (see last review, not yet fixed): The title suggests that mulit-omics techniques reveal molecular insights on GI cancers, not just help to analyse cancer.
So please add a Table 3 showing specific molecular markers of interest to diagnose or treat specifically GI cancer revealed by the modern multi-omics techiques.


Open comment 2: If possible, polish the language further with the help of a native speaker or an editing service. There are still places left where this would be helpful for the reader.

Experimental design

na

Validity of the findings

na (review)

Additional comments

see point 1

·

Basic reporting

Authors have seemingly improved the technical composition of this manuscript. Its acceptance can be recommended.

Experimental design

The study design seems to be well more outlined.

Validity of the findings

The comparative findings between spatial- and multi-omics seem to have been narrated.

---

## Round 0.3 · accepted · Accept

This manuscript is ready for publication.